# Ionic Cross-Linking as a Strategy to Modulate the Properties of Oral Mucoadhesive Microparticles Based on Polysaccharide Blends

**DOI:** 10.3390/pharmaceutics13030407

**Published:** 2021-03-19

**Authors:** Fernanda Isadora Boni, Beatriz S. F. Cury, Natália Noronha Ferreira, Maria Palmira Daflon Gremião

**Affiliations:** School of Pharmaceutical Science, São Paulo State University (UNESP), Araraquara, Road Araraquara–Jaú, Km 01, Araraquara, São Paulo 14801-902, Brazil; beatriz.cury@unesp.br (B.S.F.C.); natalia.noronha@unesp.br (N.N.F.)

**Keywords:** gellan gum, pectin, retrograded starch, mucoadhesion, liquid uptake, erosion, enzymatic degradation

## Abstract

Polymer blends of gellan gum (GG)/retrograded starch(RS) and GG/pectin (P) were cross-linked with calcium, aluminum, or both to prepare mucoadhesive microparticles as oral carriers of drugs or nano systems. Cross-linking with different cations promoted different effects on each blend, which can potentially be explored as novel strategies for modulating physical–chemical and mucoadhesive properties of microparticles. Particles exhibited spherical shapes, diameters from 888 to 1764 µm, and span index values lower than 0.5. Blends of GG:P cross-linked with aluminum resulted in smaller particles than those obtained by calcium cross-linking. GG:RS particles exhibited larger sizes, but cross-linking this blend with calcium promoted diameter reduction. The uptake rates of acid medium were lower than phosphate buffer (pH 6.8), especially GG:RS based particles cross-linked with calcium. On the other hand, particles based on GG:P cross-linked with calcium absorbed the highest volume of acid medium. The percentage of systems erosion was higher in acid medium, but apparently occurred in the outermost layer of the particle. In pH 6.8, erosion was lower, but caused expressive swelling of the matrixes. Calcium cross-linking of GG:RS promoted a significantly reduction on enzymatic degradation at both pH 1.2 and 6.8, which is a promising feature that can provide drug protection against premature degradation in the stomach. In contrast, GG:P microparticles cross-linked with calcium suffered high degradation at both pH values, an advantageous feature for quickly releasing drugs at different sites of the gastrointestinal tract. The high mucoadhesive ability of the microparticles was evidenced at both pH values, and the Freundlich parameters indicated stronger particle–mucin interactions at pH 6.8.

## 1. Introduction

The oral route is the most common pathway for drug administration, because it simultaneously provides several advantages, such as convenience, ease and security for self-administration, and improved patient compliance. However, the variation of physiological conditions exhibited throughout the gastrointestinal tract (GIT), such as pH, microbiota, enzymatic content, and peculiarities of local mucous membranes, impose great challenges for the therapeutic performance of orally administered drugs [1].

Microencapsulation can be considered a powerful technological strategy for designing innovative delivery systems for oral drug administration. This strategy allows the modulation of critical physical–chemical and/or biological properties of drug molecules, which can enhance systemic or local action depending on the formulations and technological approaches used [2,3,4,5,6]. In addition to carrying active compounds, such as drugs, nano-systems can also be microencapsulated, which may, in turn, help to effectively control drug release rates and provide protection against in vivo degradation [7,8,9,10]. One of the most attractive possibilities of employing microencapsulation in treatments is the fact that they can promote sustained, prolonged, delayed, and/or targeted release of drugs or nano-systems to specific organs or cells [8].

Over recent years, our research group has focused on the use of natural polysaccharides in the rational design of innovative micro-scale oral delivery systems, some exploiting gellan gum (GG), retrograded starch (RS), and pectin (P) as backbone materials. This approach has allowed us to modulate drug release rates and achieve desired interactions with the bio interface, influencing the biological performance of drugs [2,5,6,11,12].

GG and P are widespread, bioavailable, biocompatible, and low-cost polysaccharides. Their attractive features, such as hydrogel-forming ability, pH-dependent response, swelling, and inhibitory enzymatic activity, can be very useful in designing novel drug delivery systems. Additionally, several studies have highlighted the mucoadhesive properties of these polysaccharides [11,12].

Mucoadhesion is a complex mechanism, which is enabled by supramolecular interactions between mucous components, mainly mucin glycoproteins, and the functional groups of other substrates, such as the polymeric microparticles (PMs). The mucoadhesiveness of drug delivery systems can significantly affect the biological performance of drugs and loaded nano-systems, providing system immobilization and increasing residence time and/or absorption at the target site of action, in addition to intensifying contact with biological substrates [13].

GG, RS, and P also allow the use of mild encapsulation conditions, avoiding organic solvents, high temperatures, and extreme pH values, which preserves the stability of several drugs, proteins, and cells [14].

Blends of GG and P were exploited for the design of micro and nanostructured systems by cross-linking with Al^3+^. Both beads and nanoparticles allowed high entrapment efficiency of resveratrol and effectively reduced drug release in acidic media. Low drug permeability was also demonstrated for all cell models, revealing that such systems present promising features that allow colon-targeted drug release [2].

Retrograded starch (RS) is prepared by hydrothermally treating high-amylose starch, which increases its resistance against the enzymatic digestion in the upper portions of the GIT and enables it to be selectively degraded by colonic microbiota. Both of the aforementioned features make it suitable for designing colon-specific drug delivery systems [15,16]. Recently, a novel oral carrier for insulin, composed by GG microparticles cross-linked with aluminum (Al^3+^) and coated with films based on RS/P blends, was designed and tested. The designed microparticles effectively protected the insulin from degradation in the acidic and enzymatic conditions of the stomach, providing low drug release rates in acidic media and improving the intestinal permeability of this protein. Results evidenced the potential of this micro-carrier system for colon-specific release of proteins and other biomolecules, when aiming for systemic action [5].

The potential mucoadhesiveness of GG and RS at different pH values (1.2 and 6.8) was recently evaluated by using the rheological approach. The high mucoadhesive ability of GG was evidenced as it provoked significant changes in mucin arrangements, mainly in acidic media. On the other hand, the interactions between RS and mucin were poor, indicating low mucoadhesiveness. These findings evidence that mixing GG and RS can be a rational strategy for modulating the mucoadhesiveness of mucosal drug delivery systems to specific uses [16].

In the present work, we approach the design of inert oral mucoadhesive microparticles based on GG/RS or GG/P blends, cross-linked with calcium, aluminum, or both. Different physical–chemical properties were achieved, as well as particles mucoadhesiveness. This study opens up new possibilities for the design of inert platforms that target drugs, proteins, or even nanostructured systems to specific gastrointestinal sites.

## 2. Materials and Methods

### 2.1. Materials

Pectin (type LM-5206 CS, ~380 kDa) and gellan gum (Kelcogel® CG-LA, ~115 kDa) were kindly provided by CP Kelco (Limeira, Brazil). High-amylose starch (HAS) (Hylon VII–68% amylose, lot: HA9140) was a gift from National Starch & Chemical (New Jersey, USA). Mucin type II was acquired from Sigma-Aldrich (Missouri, USA). All other materials used were of analytical grade and obtained from commercial suppliers.

### 2.2. Methods

#### 2.2.1. Development of polysaccharide-based microparticles

##### Retrogradation of High-amylose Starch

The retrogradation process of high-amylose starch (type 3) was carried out by hydrothermal treatment, applying alternating thermal cycles of 4 °C and 30 °C every two days for 16 days, following the procedure proposed by Meneguin et al. [17]. Briefly, an aqueous high-amylose dispersion (5%, *w*/*v*) was prepared under magnetic stirring at 80 °C. This dispersion was autoclaved (121 °C) for 15 min in order to pre-gelatinize the starch before conducting the retrogradation process.

##### Polysaccharide-based Microparticles (PMs) by Ionic-cross-linking 

Microparticles of GG:P and GG:RS were prepared by the ionotropic gelation method, applying single or double ionic cross-linking with Ca^2+^ and/or Al^3+^. To summarize, GG:P microparticles were produced by mixing the polymers (1:1, *w*/*w*) and their subsequent dispersion into purified water at 2% (*w*/*v*) under magnetic stirring and heating (60 °C). Afterwards, ionic cross-linking was carried out by dripping the dispersion into the cooled cation solution (3%, *w*/*v*) using syringes with needles (22G-0.7 × 0.3 mm), under magnetic stirring. The cross-linking reaction was maintained for 30 min. In order to produce the double cross-linked microparticles, the GG:P dispersion was dripped in a Ca^2+^ solution and, after 15 min, the formed particles were separated by filtration and immersed into a cooled Al^3+^ solution under magnetic stirring for 30 min. Lastly, the obtained PMs were once again separated by filtration (reconstituted cellulose, pore size 3 µm), washed with distilled water, and dried at room temperature until reaching homogeneous weight. Particles composed of GG:RS were produced following the same protocol described above and by mixing the GG dispersion (2%, *w*/*v*) with the RS (5%, *w*/*v*) dispersion at a mass ratio of 1:2.5. The particles were filtered, washed, and dried at room temperature, reaching homogeneous weight.

#### 2.2.2. Microparticle Characterization

##### Particle Size, Span Index, and Circularity Index

Size and shape of the PMs were evaluated on a Leica MZ APO^®^ stereoscope, coupled to a Motic Images Advance 3.2 program, using captured images at 10× magnification. The circularity and the equivalent diameter of 100 particles of each sample were analyzed by the captured images using ImageJ^®^ Software.

Based on size distribution data, the Span index was determined following Equation (1), providing sample polydispersity.
(1)$$SPAN=\frac{\left(D90-D10\right)}{D50}$$

*D*90, *D*10, and *D*50 represent the diameters (μm) determined for the 90th, 10th, and 50th percentiles, respectively.

##### Surface and Internal Structure Analyses by Field Emission Scanning Electron Microscopy

Intact and fractured PMs were analyzed using high-resolution field emission scanning electron microscopy (SEM-FEG) using a JOEL-JSM-7500F microscope (Joel company, USA), coupled to the Joel Pc-100 ver.2.1.0.3. software. Samples were fixed with double-sided carbon tape and coated with a conductive material (carbon). Photomicrographs were recorded at different magnifications to enable internal and external structures visualization.

##### Liquid Uptake and Erosion of PMs

The liquid uptake profiles were determined using an Enslin device according to an adapted methodology [18]. The liquid uptake was measured at predetermined time intervals by applying different mediums, HCl 0.1 N (pH 1.2) and phosphate buffer (pH 6.8), which were considered biorelevant for GIT, simulating the gastric and intestinal environments, respectively. In pH 1.2, the assay was conducted for 120 min and, in pH 6.8, for 240 min. The liquid uptake ability of PMs was calculated according to Equation (2).
(2)$$Lu=\;\frac{Vol\;abs\;\left(mL\right)}{m\;PMs\;\left(g\right)\;}$$

*Lu* represents the media absorbed *per* mass of PMs, *V* is the volume absorbed (mL), and *m* is the initial mass of PMs (g).

##### Erosion Percentage of PMs 

PMs erosion was evaluated with the same mediums used when analyzing liquid uptake, simulating the pH value of biological fluids. A known mass of particles was placed on the mesh of an acrylic support. Thereafter, the system was immersed in HCl 0.1 N (pH 1.2) or phosphate buffer (pH 6.8) for 120 min. At pre-established time intervals, particles were removed and dried until reaching constant weight.

After drying, the PMs were weighed, and the erosion percentage was calculated according to Equation (3).
(3)$$\%E=\frac{\left(Mi-Mfs\right)}{Mfs}\times 100$$
where *%E* = percentage of erosion, *Mi* = initial PMs mass, and *Mfs* = PMs mass after drying.

In order to observe the structure of the particle matrix after the erosion test, samples were frozen (−80 °C), lyophilized for 24 h (Micromodule 115, Thermo Scientific), and analyzed using SEM-FEG.

##### Evaluation of PMs Enzymatic Degradation in Simulated Gastric and Enteric Media

The evaluation of enzymatic degradation was assessed by gravimetry and image analysis. The assay was carried out in Hanson Research dissolution equipment (ST8 Plus) equipped with apparatus I (basket), under 60 rpm agitation at 37 °C ± 0.2, using fluids that simulate the pH and enzymatic content found along the GIT [19]. A mass of particles was incubated for two hours in a NaCl 0.9% solution, acidified with HCl 0.1N (pH 1.2) containing the pepsin enzyme (0.3 mg.mL^−1^), while the other mass of particles was incubated for four hours in phosphate buffer (pH 6.8) containing the pancreatin enzyme (3.2 mg.mL^−1^) [5,20].

For gravimetric analysis, after the incubation period, samples were carefully removed from the baskets and dried at room temperature until exhibiting constant weight.

For image acquisition, samples were frozen (−80 °C), lyophilized for 24 h (Micromodule 115, Thermo Scientific, Waltham, MA, USA), and analyzed using SEM-FEG.

#### 2.2.3. Microparticle mucoadhesiveness

##### PMs Mucoadhesiveness by Porcine Mucosa Assay

Porcine intestinal mucosa was obtained from the local slaughterhouse and the ex vivo mucoadhesion was evaluated on a Texture Universal Analyzer TA.XT plus^®^ 189 (Stable Micro 190 Systems, Godalming, UK) in “compression” mode [6]. The porcine tissue was kept at room temperature and incubated in saline at 37 °C to ensure the integrity of the mucous layer. Then, sections of the mucosa were placed on the acrylic support of the equipment, and the PMs were carefully fixed to a cylindrical probe (10 mm in diameter) using double-sided tape (3M Scotch^®^) to provide the formation of a particle monolayer. The analytical probe containing the PMs was moved perpendicularly to the mucosa (5 mm.s^−1^) and introduced in the porcine intestinal mucosa (0.3 mm), remaining there during a contact time of 120 s, then, moved upward at a speed of 20 mm·min^−1^. The maximum detachment force (N) was calculated through the force vs time plots provided by the Software Texture Exponent Lite. The test was performed in triplicate (n = 3) and the results expressed as the mean and standard deviation.

##### In Vitro Mucin Adsorption 

The in vitro mucoadhesiveness evaluation of the PMs was performed according to previously described methodology [18,21]. A mass of particles (20 mg) was kept in contact with mucin aqueous solutions (Mucin type II, Sigma-Aldrich^®^) at pH 1.2 and 6.8 at different concentrations (50, 100, 150, and 200 μg.mL^−1^) and 37 °C, for one hour of incubation. Samples were centrifuged for five minutes at 3600 rpm and the free mucin at the supernatant was quantified in a spectrometer (Cary 60 UV-Vis) at 749 nm, using the Lowry colorimetric assay (Total Protein Kit, Micro Lowry, Peterson’s Modification, Sigma-Aldrich^®^). One milliliter of the Lowry reagent solution was added to 1 mL of the supernatant and kept at room temperature for 20 min. Then, 0.5 mL of the Folin-Ciocalteu’s Phenol reagent was added and kept for 30 min reaction time protected from light. A blank was obtained using purified water. Tests were performed in triplicate for each mucin concentration and pH value. Absorbance was measured and free mucin concentration was calculated applying an analytical curve previously obtained with albumin y = 0.0085x + 0.0741 (R^2^ = 0.9972). The concentration of mucin adsorbed on the PMs was determined indirectly following Equation (4).
Q _mucin adsorbed_ = Q _mucin added_ − Q _free mucin_(4)

##### Mucin Adsorption Curves

In order to investigate the mechanisms involved in the mucin adsorption process that occurs on the PMs surface, adsorption data was plotted associating free mucin concentration in the supernatant (mg.L^−1^) vs mass adsorbed to the microparticles (mg.g^−1^). Based on those adsorption curves, linearized models of Freundlich (Equation (5)) and Langmuir (Equation (6)) were applied.
(5)$$Log\;Qe=logk+\;\frac{1}{n}\;\times \;Log\;Ce$$
(6)$$\frac{Ce}{Qe}=\frac{1}{Qmáx\;b}\;+\frac{Ce}{Q\;máx}$$
where *Qe* = mass of mucin adsorbed by PMs mass; *Ce* = concentration of free mucin in the supernatant; *K* = Freundlich constant, which represents the material’s adsorption capacity; and *n* = constant of the adsorption intensity. *Qmáx* and *b* are Langmuir equation parameters, where *Qmáx* = constant of the monolayer maximum adsorption capacity and b = adsorption equilibrium constant related to the adsorption energy [22,23].

The Freundlich model was applied by plotting the graph of log *Qe* (mg.g^−1^) vs. log *Ce* (mg.L^−1^), and for the Langmuir model, the graph *Ce*/*Qe* (g.L^−1^) vs *Ce* (mg.L^−1^) was plotted. Linear regression was then applied to the acquired graphs for the acquisition of constant values. Best correlation to data was chosen based on the highest values of the coefficient of determination (R^2^).

## 3. Results and Discussion

### 3.1. Development of Polymeric Microparticles (PMs)

Microparticles based on GG:P and GG:RS blends were prepared by ionotropic gelation, using Al^3+^, Ca^2+^ or both as cross-linkers. Our aim was to investigate the effect of these different cross-linkers on the properties of the microparticles and on the modulation of their functional properties when used as oral drug delivery systems.

Samples were named according to polymer blend, concentration, and cross-linker as shown in Table 1.

For the PMs based on GG:P, the polymeric concentration of 2% (*w*/*v*) was selected, as was done in a previous study conducted by Prezotti and co-authors (2018) [21]. For GG:RS-based particles, the development followed the conditions proposed by De Oliveira (2020) [24]. The concentration of cross-linking agents was selected based on these previous works, and preliminary tests were carried out in this present study.

#### Effect of Ionic Cross-Linkers on the Size and Morphology of PMs 

Diameter control and particle size distribution are crucial because changes in volume and, consequently, on the surface area can directly affect the ability of drug or nanocarrier entrapment, resulting in potential dose variations, in addition to affecting the reproducibility of the events involved in the release process [25]. The diameters of microparticles based on GG:P and GG:RS ranged from 888 to 961 µm and from 1607 to 1764 µm, respectively, while circularity varied from 0.77 to 0.82 and from 0.76 to 0.78, respectively (Table 2).

According to Table 2, it is possible to observe that the cross-linkers significantly affected particle diameter (*p* < 0.05) in a different manner for each polymer blend. Ionic cross-linking of Ca^2+^ to GG:P resulted in bigger particles, while cross-linking with the same agent to GG:RS dispersions resulted in smaller particles. The double cross-linking process did not significantly affect the size of the particles and obtained comparable results to single cross-linking with aluminum.

Carboxylic groups of GG and P ionize in aqueous dispersions (pH ~6.0) as long as the dispersion pH remains higher than the pKa values (pKa ≈ 3.5). This fact grants high density of negative charge (~ −40 mV) to this blend. Thus, the repulsive electrostatic forces between them should provide looser and expanded polymer networks. When in contact with Ca^2+^ and/or Al^3+^ cations, extensive electrostatic interactions with the polymers’ negatively-charged functional groups are expected, which brings the chains closer and also favors other supramolecular interactions, such as hydrogen bonds and Van der Walls forces. Consequently, GG:P microparticles become characterized by a packed and stable three-dimensional network. Aluminum is a trivalent ion and has an extra positive charge in relation to the divalent calcium ion. Each Al^3+^ is able to bind to three carboxylate residues of GG and P, promoting an additional cross-linking point in relation to Ca^2+^. This behavior probably results in the formation of a more intensely cross-linked and packed structure, characterized by a smaller diameter [26], compared to particles cross-linked with Ca^2+^, which favors a more expanded network (Table 2).

RS is a polysaccharide with high molecular weight (1803 kDa) and, when compared to GG and P, in aqueous medium, it has low negative charge density (~−3mV), which makes it prone to forming weaker electrostatic interactions with the cations [27]. During GG:RS microparticle formation, it is probable that the ionic interactions between the carboxylates, from glucuronic acid molecules of GG, and Ca^2+^ and/or Al^3+^ are stronger than the ionic interactions formed with RS. Thus, the association of RS long and bulky chains should prevail by physical interpenetration throughout the interstitial spaces of the GG network. In such conditions, a more disordered and volumous structure should be built [28,29], originating larger particles than those obtained with GG:P (Table 2).

GG:RS dispersion cross-linked with calcium resulted in smaller particles (Table 2). Considering that a lower level of polymeric network cross-linking with the Ca^2+^ ions is expected, a more mobile and adaptable structure should be formed, favoring the interpenetration of RS chains and the subsequent structural rearrangements, which contribute to the formation of compact and smaller particles. On the other hand, cross-linking with Al^3 +^ resulted in a more rigid structure, especially in the outermost layer of the particle, which made the interpenetration of the RS chains more difficult, forming a more disordered and bulky structure (Table 2). Similar behavior occurred with the double cross-linking, indicating that the additional cross-linking does not affect the particle size (Table 2).

Particle size homogeneity can be quantitatively assessed by calculating the SPAN index. The lower the SPAN index value, the narrower the particle size distribution [30]. According to Table 2, all particles showed SPAN index below 0.5, highlighting size homogeneity of the obtained PMs.

The SEM photomicrographs exhibited nearly spherical PMs (Figure 1). Circularity degree analysis consists of a shape factor for which the values close to 1.0 represent a perfect circle. Herein, the nearly circular shapes of the obtained PMs was evidenced with values between 0.76 and 0.82 (Table 2).

Particles based on GG:P showed a more cohesive structure and smoother surface compared to their GG:RS counterparts (Figure 1). At the largest magnification (10,000×), the presence of cracks and fissures in the internal structure was evident for GPAl and GPCaAl (Figure 1 (E3 and F3). It is possible that during the drying process, the retraction of this more cross-linked and rigid polymeric network caused irregularities in the structure, resulting in rupture points.

Microparticles GRSCa and GRSAl presented an irregular and rough surface (Figure 1A), while the surface of GRSCaAl was more homogeneous and cohesive.

Internal images at 10,000× exhibited the presence of large pores (Figure 2D–F) in GRSCa, GRSAl, and GRSCaAl. Apparently, the double cross-linking contributed to the formation of a more cohesive matrix without the presence of these large pores and internal channels (Figure 2F). Rupture points, large pores, and internal channels can favor the diffusion of liquids into the matrix and can, consequently, accelerate the release of drugs or nano-systems.

### 3.2. Effect of Ionic cross-linkers on Liquid Uptake and Erosion of PMs in Simulated Gastric and Enteric Media

The liquid uptake ability of PMs was evaluated in media that simulate stomach and intestinal pH values (1.2 and 6.8, respectively).

The profiles of absorbed liquid (mL) vs mass of PMs (g) were presented in Figure 3. The amount of liquid uptake is directly related to the hydrophilicity of the polymeric blend used in the matrix and to its cross-linking degree. The former not only affects the hydrophilicity but also impacts packing and arrangement of the polymer matrix [23,24,31].

In the acid medium (pH 1.2), differences in absorption rates were observed, which may have been influenced by the particle composition and by the cross-linker. Microparticles composed by GG:RS absorbed similar volumes of acid medium at the end of the test; however, the absorption rate varied between samples. Fifteen minutes into the test, GRSCa absorbed 0.73 mL.g^−1^, while GRSAl and GRSCaAl absorbed a volume almost two times greater (~ 1.35 mL.g^−1^). In 30 min, GRSCa absorbed 1.31 mL.g^−1^, while GRSAl and GRSCaAl absorbed 1.62 and 1.73 mL.g^−1^, respectively. After 60 min, the difference observed was maintained, with the samples of GRSCa, GRSAl, and GRSCaAl absorbing 1.85, 2.15, and 2.17 mL.g^−1^, respectively. After 120 min, samples had absorbed the same final volume (Figure 3).

The same behavior was observed for particles composed by GG:P. After 15 min of testing, the GPCa particle had absorbed 0.95 mL.g^−1^ of the medium, value higher than that observed for GPAl and GPCaAL, which absorbed 0.75 and 0.54 mL.g^−1^, respectively. At 30 min, the volumes absorbed by the samples were similar (~ 1.5 mL.g^−1^) but differentiating again at 60 min with GPCa, GPAl, and GPCaAl absorbing 2.75, 2.16, and 1.93 mL.g^−1^, respectively. At the end of the test, the significant difference remained with the highest volume of acid medium being absorbed by GPCa (4.01 mL.g^−1^) (Figure 3).

When analyzing the influence of the cross-linker in each polymer blend, we observed that GG:RS formed a packaged matrix with no visible pores and cracks when calcium was used as the cross-linker. This made the acid medium diffusion difficult, resulting in the lowest absorption rate among all PMs. GG:RS particles, formed by double cross-linking, presented similar behavior to those cross-linked only with Al^3+^, which may indicate that the organization of the polymeric net, initially cross-linked with Ca^2+^, did not undergo significant changes after contact with the second cation. Another possibility is that the changes in the structure occurred more in the outer region of the particle.

In contrast, cross-linking GG:P with Ca^2+^ promoted a lower number of ionic bonds between the carboxylates of the GG and P chains, forming a matrix with a lower degree of cross-linking and consequently greater hydrophilia. This PMs was also composed by a network with greater mobility, capable of accelerating the diffusion of the medium and more easily accommodate a large volume of liquid in its structure.

In acid medium, GG:P particles exhibited a higher percentage of erosion, with values ranging from 20.1 to 26.4% (Figure 3). In this case, the cross-linker used did not have a significant impact on results. At this pH, the polymer network probably remained more packed because of the protonation of carboxylic groups of polymers. Thus, the volume of medium that was diffused into the matrix was capable of exerting sufficient hydrostatic pressure to disrupt the rigid polymeric structure, causing ruptures that resulted in the accelerated erosion of the system [21].

The photomicrographs (Figure 3) revealed that, although the particles maintained a spherical shape, they exhibited a certain degree of superficial erosion, characterized by decreased size and superficial heterogeneity with roughness and fissures.

The image of the transversal sections of these particles allows us also to observe that the internal structure was compact and homogeneous, indicating that the excess of H^+^ ions from the acidic media enabled the protonation of free carboxylic groups of the polymeric structure, reducing the repulsion of the polymer chains. Thus, the interactions by hydrogen bonds and dipole-induced interactions are favored and a more cohesive structure is built, which can hamper the diffusion of the medium into the matrix, resulting in erosion mostly of the superficial layer of the particles.

In phosphate buffer (pH 6.8) for up to 60 min, the observed profiles were similar to those obtained in an acid medium. After 120 min, buffer absorption became higher than in acid medium for all samples (up to 1.4 times). From 120 to 180 and to 240 min, an increasing uptake of phosphate buffer was observed, reaching values ranging from 5.17 to 7.71 mL.g^−1^ (Figure 3).

Particles GRSAl and GRSCaAl absorbed the highest volumes of phosphate buffer, 7.72 and 7.27 mL.g^−1^, respectively, after 240 min. In contrast, GRSCa, GPCa, GPAl, and GPCaAl all absorbed similar volumes (Figure 3). This was probably due to the less extensive cross-linking of GG with Ca^2+^. The RS chains were able to interpenetrate more efficiently throughout the polymer meshes, accommodating its chains in a more packed way. This structural arrangement probably hindered the expansion of the matrix by electrostatic repulsion and the diffusion of liquids into the structure, resulting in lower liquid uptake.

Considering that liquid absorption represents the first stage of the drug or nanocarriers release process, the higher absorption of phosphate buffer (pH 6.8) should increase the relaxation and mobility of the polymer chains, which can favor the diffusion of the drug or nanocarriers from the particle into the intestinal environment.

In pH 6.8, erosion was lower than in acidic pH, with the exception of sample GRSCaAl, which presented a similar percentage of erosion in both mediums, 18.7% and 20.1%, respectively (Figure 3). At pH values higher than the polymer pKa, free carboxylic acid groups were ionized, which resulted in the electrostatic repulsion of the chains and in a more dilated and mobile structure [4]. With the expansion of the polymer meshes, the matrix was able to accommodate a greater number of water molecules, suffering less erosion, but exhibiting a higher degree of swelling [21]. This behavior was corroborated by the photomicrographs (Figure 3), in which the tendency to maintain the shape of the particles was observed, but with a significant enlargement of the particle, indicating the swelling process. The internal structure of PMs at pH 6.8 demonstrated the formation of a greater number of channels and pores, resulting in a laminated and dilated structure, which favors the release of drugs or nanocarriers.

### 3.3. Effect of Ionic Cross-Linkers on Enzymatic Degradation of Mps in Simulated Gastric and Enteric Media

The enzymatic degradation (%) of PMs are shown in Table 3. Microparticle degradation in simulated gastric media (containing pepsin, pH 1.2) ranged from 18.2 to 61.3% and, in simulated enteric media (containing pancreatin, pH 6.8), from 15.1 to 95.3%.

Digestive enzymes are produced and secreted by the mouth, stomach, pancreas, vesicle, and intestines and help in the digestion process so that, the constituents of substances consumed become bioavailable. Pepsin originates in the stomach and its activation occurs through the presence of HCl. Pancreatin, on the other hand, is composed of a set of enzymes, such as amylase, trypsin, lipase, and pancrease, which have the ability to digest proteins, polypeptides, and starch [31].

According to Table 3, GG:RS microparticles cross-linked with calcium enabled an expressive reduction of enzymatic degradation in both media, behavior that probably willresult in an extended release profile. As previously described, the cross-linking of this blend with calcium promoted the formation of a compact network, improving its resistance against the enzyme access and degradation. It was also observed that GG:RS microparticles were more resistant than GG:P micoparticles, probably due to the retrogradation of starch, which reduced the access of enzymes to the polymer matrix.

In acidic media, calcium cross-linking significantly increased the enzymatic degradation of GG:P microparticles, which contrasts GG:RS. This is in agreement with our findings about the formation of a more flexible and mobile network, which favors enzyme assessment, while Al^3+^ and double cross-linking form a more rigid and voluminous structure.

In simulated enteric media, GPAl was the sample that suffered the greatest degree of degradation, followed by the GPCaAl, with 95.3% and 75.2%, respectively. This is an interesting behavior if a burst release in the intestine is desired. The GG:P matrix granted easier access to enzymes, since at pH 6.8 there was an expansion of the polymeric net formed by GG:P (pKa ~ 3.0), resulting in the high rates of matrix swelling and enzyme diffusion to the interior of the microparticle.

Afterwards, samples were collected and analyzed by microscopy (Figure 4).

It is possible to observe that in the simulated gastric medium, most samples exhibited preserved structures, maintaining their nearly spherical shape. In the simulated enteric medium, however, structures exhibited deformations or even disintegration, especially microparticles: GRSAl, GRSCaAl, GPAl, and GPCaAl.

In comparison with the previous test (item 3.2), the addition of enzymes significantly affected the structural integrity of the microparticles, mainly in the conditions that simulate the enteric medium, revealing their pH/enzymatic sensitivity behaviors.

### 3.4. Effect of Ionic Cross-linking on Mucoadhesiveness

#### 3.4.1. PMs Mucoadhesiveness by the Porcine Mucosa Assay

This experiment used intestinal porcine tissue as the biological substrate that mimicked the physiological conditions to which the system would be exposed. Maximum mucoadhesion force (*FMax*), also known as potential mucoadhesiveness, was evaluated by measuring the strength required to detach microparticles from the substrate.

The average *FMax* of raw RS and GG polymers were 0.36 N and 1.23 N, respectively, values higher than that of P (0.12 N, Figure 5).

GG is a hydrophilic polymer that contains several carboxylic and hydroxyl groups in its chain. These groups can interact with the glycoproteins present in the mucus via supramolecular interactions, such as hydrogen bonds, which form extensive interactions of considerable strength and, consequently, increase the force needed for tissue detachment [11,12].

RS presented higher *FMax* than P, and according to diffusion theory, this is probably due to its extensive and flexible chains, which allow greater surface contact with the mucosa, favoring the interpenetration of the mucus layer [13,16].

The P used in this study was of low molecular weight and low degree of esterification (DE < 50%). Other studies have reported that low DE pectin has the capacity of penetrating deeply into the intestinal mucus layer, but is unable to adhere strongly to the surface [32,33]. The low molecular weight of P probably provides less folding and diffusion in the mucus layer, resulting in lower *FMax* (Figure 5).

Particles presented lower *FMax* than raw polymers GG and AR, which can be explained by the structural rearrangement of the chains that occurred because of the cross-linking process. On the other hand, the particles obtained from the GG:P mixture presented higher *FMax* than the isolated P, highlighting the importance of GG association for improving system mucoadhesiveness.

Cross-linking GG:RS with aluminum favored mucoadhesiveness, while cross-linking GG:P with calcium caused this same effect. Thus, GRSAl and GPCa were the samples that presented the highest *FMax* values (0.21 N and 0.18 N, respectively), and the GRSCa and GPAl presented the lowest *FMax* (0.10 N).

The more flexible and mobile structure of GPCa should make its chains more available for interpenetration with mucin chains and subsequent supramolecular interactions. Despite the more rigid and packed structure exhibited by GRSAl, the higher mucoadhesive ability of both polymers in relation to P probably contributes to this behavior.

Double cross-linking did not provide significant changes in mucoadhesiveness in Ca^2+^ cross-linked GG:RS microparticles and Al^3+^ cross-linked GG:P microparticles.

#### 3.4.2. In vitro mucin absorption

GRSCa and GPCa PMs, which exhibited the lowest and highest mucoadhesive strength, respectively, were selected for the study of mucin adsorption.

Mucin adsorption curves were used to comprehend the mechanisms that drive the mucoadhesion of these microparticles.

Figure 6 shows the graph of absorbed mucin (mass) and percentage of mucin adsorbed in the PMs, according to mucin concentration in the aqueous solutions at pH 1.2 and 6.8. The adsorption percentages on microparticle surfaces varied from 84% to 97% in the highest concentration of mucin at both pH values. In both media, the mucin adsorption increased as the concentration of available mucin increased, while the polymer blend did not significantly influence the amount of adsorbed mucin (Figure 6).

From the mucin adsorption data, curves that represent the interaction between mucin and the particle surfaces were plotted (Figure 7). In these curves, the mass of mucin adsorbed by microparticle mass (Qe, mg.g^−1^) was associated to the concentration of free mucin in the supernatant (Ce, mg.L^−1^).

The profiles indicate favorable interaction, in which the amount of adsorbed mucin increases as the concentration of mucin in the medium increases. Despite the small difference in the results of adsorbed mucin mass, the greater slope exhibited by the mucin adsorption curves at pH 1.2 suggests the greater adsorption ability of microparticles in this medium.

Figure 7 shows the differences between adsorption curve profiles according to pH variation. According to the classification of Giles et al. [23], the GPCa curve, at both pH values, belongs to the “L” class, which indicates that the availability of active sites for adsorption on the particle surface decreases as the concentration of mucin in the medium increases. In this curve, the inflection point characterizes the saturation of the interaction sites on the particle surface. However, at higher concentrations, the adsorption process continues, with the mucin binding to sites that are energetically different from those initially saturated [34]. The same “L” class was fitted with the mucin adsorption data of GRSCa at pH 6.8, however, at the concentrations evaluated, the curve did not show a plateau that would indicate a limit on the adsorption capacity.

At pH 1.2, the GRSCa adsorption curve belongs to the “S” class, with an upward curvature, since particle-mucin interactions were probably weaker than the mucin–mucin interactions [23].

From the values of R^2^ obtained through linear regression, it was observed that the adsorption data shows better correlation with the Freundlich model (at pH 1.2, R^2^ = 0.9835 and 0.9843, and at pH 6.8, R^2^ = 0.9849 and 0.9995, for GPCa and GRsCa, respectively).

The Freundlich model describes adsorption onto an irregular surface, as well as the possibility of adsorption in multiple layers. In this model, the amount of adsorbed solute results from the sum of adsorption in all available sites, each with different binding energy, with the strongest binding sites being occupied first, followed by adsorption in the lower energy sites, until reaching the process balance.

From the linearization of the Freundlich model, the coefficients *n* and *k* were obtained, with *n* indicating the intensity of mucin adsorption on the particle surface. At pH 1.2, the values of *n* were 1.56 and 1.38, and at pH 6.8, they were 1.76 and 1.73 for GRSCa and GPCa, respectively. Favorable conditions for adsorption occur when values of *n* are greater than 1.0.

The *k* coefficient, on the other hand, is related to particle adsorption capacity. At pH 1.2, *k* = 8.5 and 7.3, and at pH 6.8, *k* = 10.1 and 9.6 for GRSCa and GPCa, respectively. The high values of *k* indicate the great capacity of mucin adsorption per unit of PMs mass.

In the tested range of mucin concentration, the adsorption was favorable at both pH values; however, the adsorption coefficients *n* and *K* of the Freundlich model indicate that the particle–mucin interaction was strongest at pH 6.8.

## 4. Conclusions

In this work, microparticles based on different polymer blends (GG:RS and GG:P) were prepared by ionotropic gelation using different cations (Ca^2+^ or Al^3+^), or by double cross-linking, in order to modulate both their physical–chemical and mucoadhesive properties. Our intent was to design mucoadhesive oral carriers for targeted delivery of drugs or nano systems to different sites of the GIT. The cross-linking process with calcium or aluminum promoted different effects on the GG:RS and GG:P properties, representing a promising strategy for their modulation according to specific purposes. In general, double cross-linking did not promote significant changes, especially when compared to aluminum cross-linking.

Calcium cross-linking promoted the decrease of GG:RS microparticles size, while the same process done on GG:P caused the contrary effect. The liquid uptake ability of both GG:RS and GG:P microparticles was high, with GRSCa and GPCa presenting the lowest and highest acid absorption rates, respectively. In phosphate buffer, particles absorbed high volume per mass, and the different cross-linking approaches did not affect this behavior. Cross-linking of GG:RS microparticles with calcium resulted in an impressive reduction of microparticle degradation in mediums containing enzymes and of pH 1.2 and 6.8, a favorable feature for protecting drugs against premature release in acidic media and a sustaining drug release in the intestine. In contrast, GG:P microparticle degradation was significantly increased by cross-linking with calcium at both pH media, which can provide quick release of drugs throughout the GIT. Aluminum cross-linking significantly increased microparticle degradation in simulated intestinal media, which can contribute to fast release of drugs in this site.

The ex vivo and in vitro tests evidenced the mucoadhesive ability of GG:RS and GG:P cross-linked microparticles, regardless of the pH. This constitutes a promising feature for the immobilization of the drugs at different sites of action and/or absorption. Calcium cross-linking enhanced the mucoadhesiveness of GG:P microparticles while aluminum hampered mucoadhesiveness in GG:RS blends.

The promising attributes of the inert microparticles designed in this work reveal their potential for encapsulation different drugs or nanoparticles, aiming the targeted release at different sites of the GIT. The effect of the encapsulation of nanoparticles on the microparticles structure and properties will be addressed in future work.

## Figures and Tables

**Figure 1 pharmaceutics-13-00407-f001:**
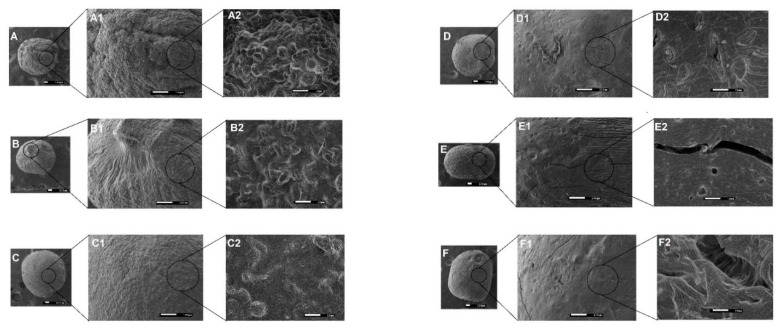
Photomicrographs of the polymeric microparticles (PMs) surface (**A**: GRSCa at 40×, **A1**: 150×, and **A2**: 1500×; **B**: GRSAl at 40×, **B1**: 150×, and **B2**: 1500×; **C**: GRSCaAl at 40×, **C1**: 150×, and **C2**: 1500×; **D**: GPCa in 40×, **D1**: 150×, and **D2**: 1500×; **E**: GPAl in 40×, **E1**: 150×, and **E2**: 1500×; and **F**: GPCaAl in 40×, **F1**: 150×, and **F2**: 1500×).

**Figure 2 pharmaceutics-13-00407-f002:**
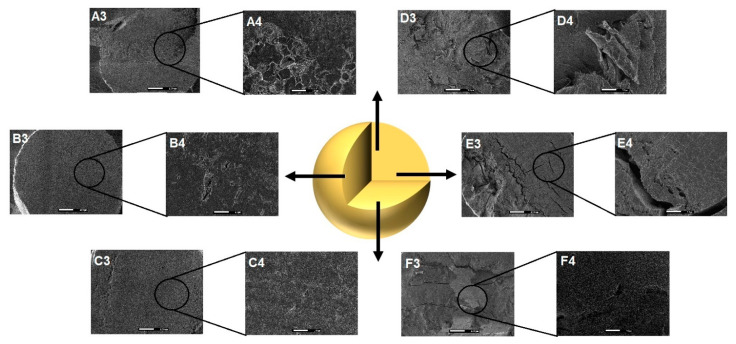
Photomicrographs showing the internal structure of the PMs (**A3**: GRSCa 150×, and **A4**: 1500×; **B3**: GRSAl 150×, and **B4**:1500×; **C3**: GRSCaAl 150×, and **C4**:1500×; **D3**: GPCa 150×, and **D4**:1500×; **E3**: GPAl 150×, and **E4**: 1500×; and **F3**: GPCaAl 150×, and **F4**: 1500×).

**Figure 3 pharmaceutics-13-00407-f003:**
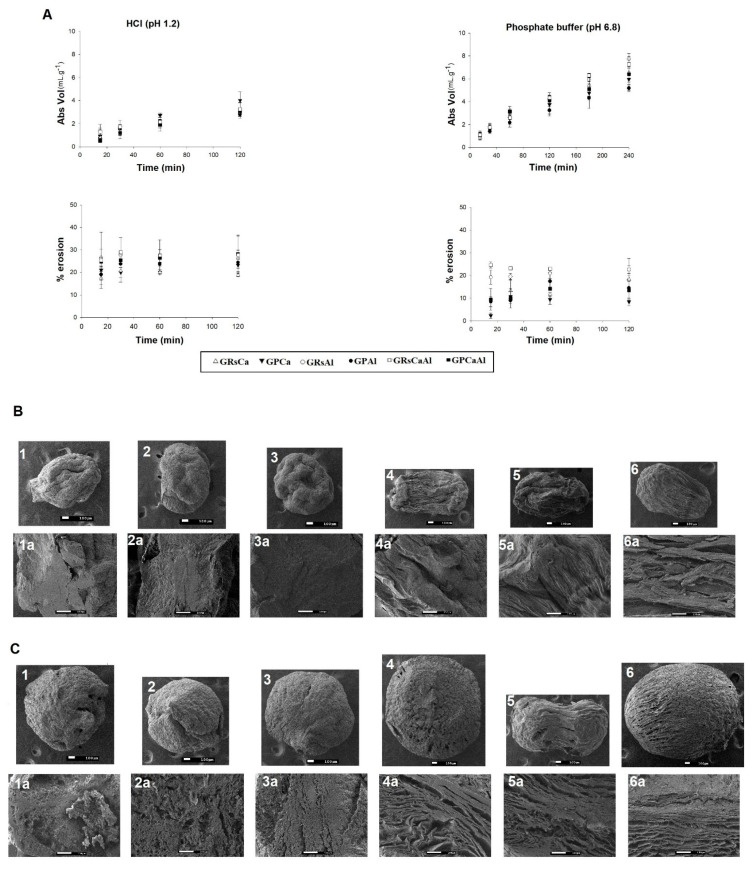
**A**- Samples profiles of liquid uptake and erosion; **B**-Photomicrographs of PMs after incubation (120 min) in HCl (pH 1.2); **C**- Photomicrographs of PMs after incubation (120 min) in phosphate buffer (pH 6.8). (1: GRSCa 40×; 1a: GRSCa 150×, 2: GRSAl 40×; 2a: GRSAl 150×, 3: GRSCaAl 40×; 3a: GRSCaAl 150×, 4: GPCa 40×; 4a: GPCa 150×, 5: GPAl 40×; 5a: GPAl 150×. 6: GPCaAl 40×; 6a: GPCaAl 150×).

**Figure 4 pharmaceutics-13-00407-f004:**
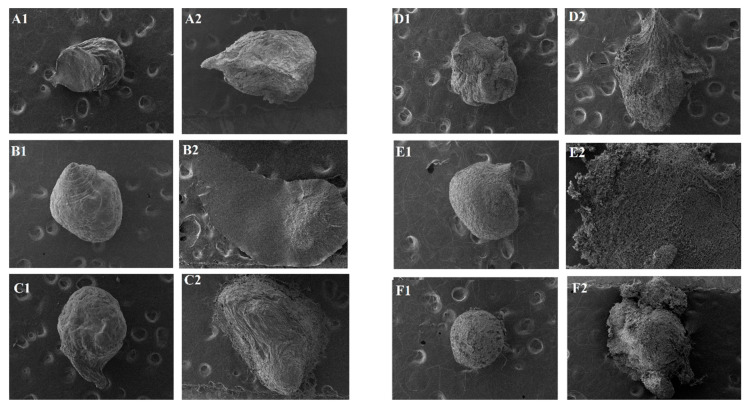
Photomicrographs of microparticles after enzyme degradation assay (40×)- **A**: GRSCa; **B**: GRSAl; **C**: GRSCaAl; **D**: GPCa; **E**: GPAl, and **F**: GPCaAl, 1: gastric medium and 2: enteric medium).

**Figure 5 pharmaceutics-13-00407-f005:**
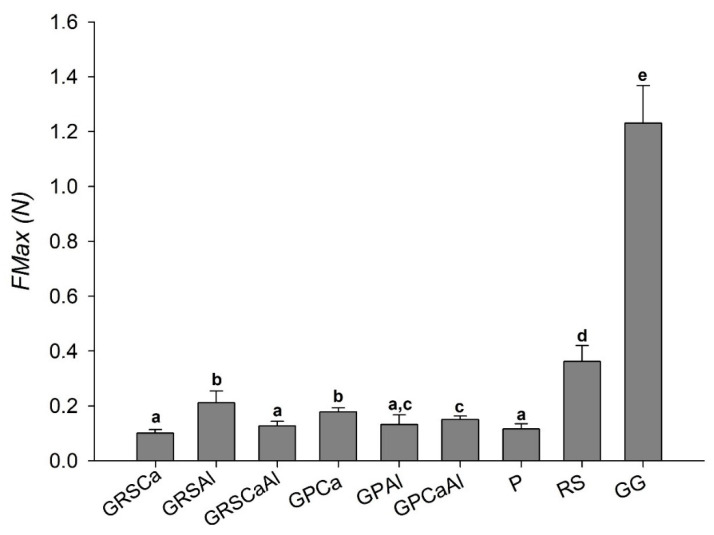
Maximum mucoadhesion force (*FMax*) of microparticles: pectin (P), retrograded starch (RS), and gellan gum (GG).

**Figure 6 pharmaceutics-13-00407-f006:**
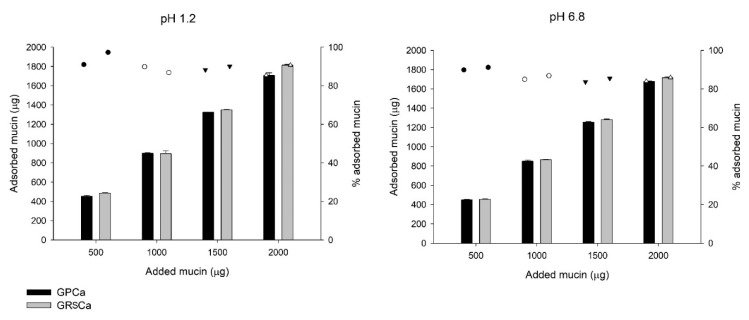
Amount of mucin adsorbed by the microparticles as a function of the amount of mucin added to the solutions (represented by the bars) and percentage of adsorbed mucin (represented by the dots).

**Figure 7 pharmaceutics-13-00407-f007:**
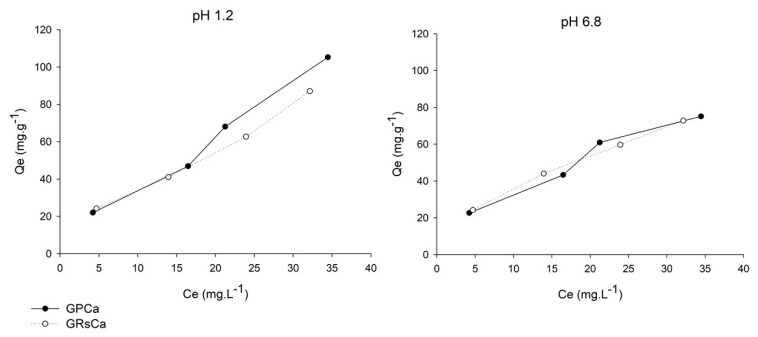
Mucin adsorption curves of microparticles in mediums of pH 1.2 and 6.8.

**Table 1 pharmaceutics-13-00407-t001:** Microparticle nomenclature and composition.

Sample	Mass Ratio of GG:P (*w*/*w*)	Mass Ratio between GG:RS (*w*/*w*)	Cross-linker (%)
GRsCa	-	1:2.5	Ca^2+^ (3%)
GRsAl	-	1:2.5	Al^3+^ (3%)
GRsCaAl	-	1:2.5	Ca^2+^ e Al^3+^ (3%)
GPCa	1:1	-	Ca^2+^ (3%)
GPAl	1:1	-	Al^3+^ (3%)
GPCaAl	1:1	-	Ca^2+^ e Al^3+^ (3%)

**Table 2 pharmaceutics-13-00407-t002:** Microparticle average diameter, circularity, and SPAN index (n = 100).

Sample	Average Diameter (µm) ± SD	Circularity ± SD	SPAN
GRsCa	1607 ± 177	0.76 ± 0.07	0.29
GRsAl	1793 ± 158	0.78 ± 0.09	0.30
GRsCaAl	1760 ± 174	0.78 ± 0.09	0.28
GPCa	961 ± 97	0.77 ± 0.07	0.27
GPAl	889 ± 110	0.82 ± 0.05	0.41
GPCaAl	888 ± 99	0.81 ± 0.08	0.30

**Table 3 pharmaceutics-13-00407-t003:** Enzymatic degradation of the GG:P (GP) and GG:RS (GRS) microparticles cross-linked with Ca^2+^ (Ca), Al^3+^ (Al), or Ca^2+^ and Al^3+^ (CaAl).

Sample	HCl+NaCl (pH 1.2) with Pepsin	Phosphate Buffer (pH 6.8) with Pancreatin
	Degradation (%) ± SD	
GRSCa	18.2 ± 0.6	15.1 ± 9.2
GRSAl	32.0 ± 0.8	53.7 ± 7.5
GRSCaAl	40.6 ± 2.1	52.9 ± 3.8
GPCa	61.3 ± 0.7	65.7 ± 1.8
GPAl	35.8 ± 2.8	95.3± 3.7
GPCaAl	32.1 ± 0.1	75.2 ± 1.5

## Data Availability

The data presented in this study are available on request from the corresponding author.

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
