# Peer review of "Ionic Cross-Linking as a Strategy to Modulate the Properties of Oral Mucoadhesive Microparticles Based on Polysaccharide Blends"

_pharmaceutics, 2021, doi:10.3390/pharmaceutics13030407_

Round 1

Reviewer 1 Report

The manuscript “Ionic cross-linking as a strategy to modulate the properties of oral mucoadhesive microparticles based on polysaccharide blends” by Boni et al. is well written and easy to understand for the reviewer. The manuscript describes the formulation of mucoadhesive microparticles made by polymer blends (i.e. gellam gum, resistant starch and pectin) cross-linked with calcium or aluminium. The authors obtained microparticles of different sizes depending on the polymer blend and the ion used, and they extensively characterized the particles in terms of rate of medium uptake and erosion at pHs simulating the gastrointestinal tract (1.2 and 6.8). Eventually, the microparticles were investigated for their mucoadhesive properties at both pHs.

The reviewer suggests a careful English editing. The verb is missing in some sentences (e.g.: line 561-562) and the use of certain words is inappropriate completely changing the meaning.

Major revisions

Although the results are clearly presented and the study is interesting, there is a major comment to be solved.

The authors are proposing the obtained particles as drug delivery systems with specific features, however, they must provide evidence of the systems capacity to effectively and quantitatively load at least a model drug to support the proposed study. The presence of a drug could alter the structure of the formed microparticles affecting their properties, and this must be investigated.

Minor revisions

  • Details about polymers molecular weight could be added to the materials section.
  • Line 28: expressive? Probably it was meant to be impressive.
  • Line 129: the authors mentioned they have filtered the polymer mixture after the cross-linking reaction? Could you explain why?
  • Line 130: microparticles were filtered to be isolated, please specify the filter pore size and material.
  • Line 161. Eq 2: please remove the bracket before Vol abs.
  • Line 178: en teric media should be enteric media
  • Line 180: Nor in the method neither in the results and discussion there is a clear explanation on how the enzymes concentration were selected to conduct the degradation experiments. Please motivate your choice or add appropriate references.
  • Line 184: HCL : correct with HCl
  • Line 186: Particles was incubated for four hours or in a phosphate buffer: remover or
  • Line 225. Evolved? Did the authors mean involved?
  • Figure 1: the sizes of the pictures are different in the same magnification range (for instance, all images 40x are of different sizes). The reviewer strongly suggest to make all images sizes homogeneous per magnification. Scale bars are missing.
  • Figure 3: same comment as above. In addition, there are no indication of the magnifications used for the pictures. Moreover, it would be easier for the reader if letters will be added to distinguish the different panels and details are added to the Figure caption. Also, please give an explanation about the fact that all studies were conducted for 120 minutes a part from the liquid uptake at pH 6.8.
  • Line 443: packaged?
  • In the in vitro mucin absorption assay the authors declared that being Ca2+ a safer cross-linker they performed the following test using only this ion. What is the rationale behind the previous assay conducted with Al3+, then?
  • Line 537: the authors indicated the presence of a L shape for the curves reported in Figure 7. The reviewer quite disagree with this consideration and can’t identify this L shape. Please comment on that.

Reviewer 2 Report

In the manuscript, pharmaceutics-1115162 microparticles based on different polymer blends were prepared by ionotropic gelation using a single or combination of crosslinking agents. The microparticles were characterized and the effect of crosslinking agent evaluated. 

The manuscript is well written in all the part; the aim of the work is clear and the results are well described, even if in some parts should be supported by references.

Even if there is no novelty in the reported work, I believe it could be interesting for the readers of pharmaceuticals which are working in the microencapsulation field. 

However, I do not suggest for publication in the present state as some points have to be clarified:

1) In the preparation of the microparticles the use of different crosslinking agents or a combination of the crosslinking agent were investigated. The authors report a concentration of 3% of crosslinking agent. On which base it was chosen? in case of a combination of Al+3 and Ca+2, how was considered the competition between the two ions? 

2) Figure 1,2,3. Scale bar should be indicated in the micrographs

3) Figure 3. The SEM micrographs represent the particles in different media. To which time they refer? It would be interesting to report micrographs of the particles at the beginning and at the end of the exposition time.

4)Section 3.2..The authors should support their statements related to the erosion and different effect of the crosslinking agent with references. 

5) Alginic acid is a widely used polysaccharide which easily interacts with Ca+2 to give nanoparticles, microparticles, hydrogel and so on. What are the advantages of using the mentioned polysaccharides instead of alginic acid?

6) In the preparation of the microparticles Ca+2 and Al+3 are used. Which can be the effect on the physiological environment, in particular on the enzymatic activity, once Ca+2 and Al+3 are released?

7) why the authors did not try to load a bioactive compound which has a therapeutic effect in the stomach of in the intestine? It could be interesting to evaluate the impact of drug loading on the properties of the particles and in particular in the crosslinking.

8) Keep the same font and size in the whole manuscript

Round 2

Reviewer 1 Report

Despite the fact that all the minor revisions have been addressed by the authors, the major comment still remains unsolved. In the present form, the manuscript lack to demonstrate the capacity of the proposed technology to be suitable for drug loading and delivery, which is the major scope of the work.

Author Response

We understand the Reviewer's question. Preliminary studies by the group demonstrated the capacity of this type of system in the incorporation (between 20 to 85%) of drugs such as ketoprofen, resveratrol and methotrexate (data not shown). However, the aim of this work was the development and characterization of mucoadhesive platforms as oral carriers for drugs or nanoparticles. The platforms were well characterized in terms of mucoadhesion, degree of crosslinking, morphology, erosion and medium uptake. In this context, at the ‘Conclusion’ of the manuscript, it was highlighted that future studies will be published, addressing the effect of drugs or nanoparticles incorporation on these platforms, as well as the biological behavior of these systems (page 18).

Reviewer 2 Report

The authors answered all the comments and improved the manuscript.

Author Response

We thank the Reviewer for the interest in our work, as well as the time taken for the careful revision.